# Rhizoferrin Glycosylation in *Rhizopus microsporus*

**DOI:** 10.3390/jof6020089

**Published:** 2020-06-18

**Authors:** Anton Škríba, Rutuja Hiraji Patil, Petr Hubáček, Radim Dobiáš, Andrea Palyzová, Helena Marešová, Tomáš Pluháček, Vladimír Havlíček

**Affiliations:** 1Institute of Microbiology of the Czech Academy of Sciences, Vídeňská 1083, 142 20 Prague, Czech Republic; anton.skriba@biomed.cas.cz (A.Š.); rutuja.patil@biomed.cas.cz (R.H.P.); palyzova@biomed.cas.cz (A.P.); maresova@biomed.cas.cz (H.M.); tomas.pluhacek@biomed.cas.cz (T.P.); 2Department of Analytical Chemistry, Faculty of Science, Palacký University, 771 46 Olomouc, Czech Republic; 3Department of Medical Microbiology, 2nd Faculty of Medicine, Charles University and Motol University Hospital, 150 06 Prague, Czech Republic; petr.hubacek@fnmotol.cz; 4Public Health Institute in Ostrava, 702 00 Ostrava, Czech Republic; radim.dobias@zuova.cz

**Keywords:** *Rhizopus microsporus*, glycoside, rhizoferrin, metabolite, siderophore, mass spectrometry, liquid chromatography, human isolate, posaconazole metabolism

## Abstract

*Rhizopus* spp. are the most common etiological agents of mucormycosis, causing over 90% mortality in disseminated infections. The diagnosis relies on histopathology, culture, and/or polymerase chain reaction. For the first time, the glycosylation of rhizoferrin (RHF) was described in a *Rhizopus microsporus* clinical isolate by liquid chromatography and accurate tandem mass spectrometry. The fermentation broth lyophilizate contained 345.3 ± 13.5, 1.2 ± 0.03, and 0.03 ± 0.002 mg/g of RHF, imido-RHF, and bis-imido-RHF, respectively. Despite a considerable RHF secretion rate, we did not obtain conclusive RHF detection from a patient with disseminated mucormycosis caused by the same *R. microsporus* strain. We hypothesize that parallel antimycotic therapy, RHF biotransformation, and metabolism compromised the analysis. On the other hand, the full profile of posaconazole metabolites was retrieved by our in house software CycloBranch.

## 1. Introduction

Mucorales are mainly saprophytes that are commonly found in soil and decomposing material. Nevertheless, some of these fungi have been used for centuries in food manufacturing for cheese ripening or Asian fermented food production. The metabolome of mucoromycetes is poorly studied, and only a limited number of reports have appeared so far [1,2]. Fifteen different sterols were detected in clinically relevant Mucorales species in a targeted metabolomics fashion, after derivatization, by gas chromatography-mass spectrometry [3]. In an older report, trisporic acids were described as C-18 terpenoid pheromone compounds responsible for sexual differentiation in Mucorales [4]. Trisporic acid E, along with 12 other apocarotenoids, was identified in the culture media of *Phycomyces blakesleeanus.* Cytotoxic 3-nitropropionic acid was detected in the culture extract of *Mucor circinelloides* [5].

Certain *Rhizopus microsporus* strains have erroneously been reported to produce either pharmaceutically active rhizoxins or highly toxic rhizonins [6]. In a consequent study, other authors demonstrated that these metabolites are not secreted by the fungus but are produced by *Bulkholderia* symbionts that reside within the fungal cytosol [7]. As *R. microsporus* has been used for fermented soy production, one cannot exclude a link between intracellular bacteria producing toxins and their harmful effects on humans. The small molecule metabolome of selected Mucorales species is summarized in Table 1.

All mucoralean genomes available in 2014 were reported to lack nonribosomal peptide synthetases (NRPSs) [17]. Controversially, more recent work identified polyketide synthase, NRPS, and L-tryptophan dimethylallyl transferase encoding genes in Mucoromycota [2]. Mucorales also synthesize polycarboxylate siderophores, including rhizoferrin (RHF) [18], which have a much weaker binding activity than hydroxamate siderophores but may be important for microbial virulence, as reported, e.g., in *Francisella tularensis* [19]. Recently, rhizoferrin biosynthetic genes have been identified in the Mucorales species *R. delemar* [9]. An expressed and purified 72 kDa protein Rfs corresponded to a superfamily of adenylating enzymes, and its sequence was similar to that of a homologous bacterial NRPS-independent siderophore (NIS) protein, SfnaD, that is involved in the biosynthesis of staphyloferrin A in *Staphylococcus aureus* [20]. 

Siderophores represent a subset of metallophores, a diverse family of secondary metabolites with numerous biological functions covering resistance to reactive oxygen species, ability to produce sexual spores, and modulation of host functions; these compounds can be both genotoxic and/or cancerostatic [21]. Metallophores are metal-chelating molecules secreted by pathogens to facilitate metal acquisition/passivation, thereby representing a significant group of microbial pleiotropic virulence factors [22]. These molecules have a high secretion rate at the invasive stage of a disease and can be strain specific. They are used as therapeutics [23,24], and have also been defined as critical early markers of fungal infections [25,26]. The complex structures of siderophores offer great diagnostic potential in terms of specificity, but variable or mixed moieties confer the final physicochemical properties that affect sample preparation [27]. In contrast to immune host responses to the presence of infections, which are spatiotemporally nonspecific, direct monitoring of microbial siderophores is specific. False positivity is almost excluded, as siderophores are not synthesized by mammalian cells except for small catechols [28,29].

Specifically, for Mucorales, RHF was reported in *Syncephalastrum racemosum*, *Rhizomucor pusillus*, *Mucor hiemalis* [9], a number of arbuscular mycorrhizal fungi [15], and the other Mucorales species listed in Table 1. RHF is also actively secreted by some clinically important bacteria, including *Francisella tularensis* [19], *Ralstonia (Pseudomonas) pickettii* [30], and *Legionella pneumophila* (in [31] referred to as legiobactin). As a xenosiderophore, RHF uptake has been described in *Morganella morganii* [32] and *Mycobacterium smegmatis* [33]. Some RHF analogs were prepared by directed fermentation [14].

Importantly, Mucorales genera *Rhizopus*, *Lichtheimia* (formerly *Absidia*), *Mucor*, *Rhizomucor*, and *Cunninghamella* have been isolated in high abundance from patient material. The major obstacle in the management of mucormycosis has been a lack of a noninvasive, rapid, and reliable diagnostic test [34]. *Rhizopus* spp. are the most common etiological agents of mucormycosis, causing over 90% mortality in disseminated infections [35]. The high rate could be attributed to late invasive sampling from primarily sterile material and inadequate activity of antifungals applied in prophylaxis [36].

Of note, the bacterial siderophore deferoxamine B (Desferal) can be misappropriated by Mucorales to stimulate their growth in patients with mucormycosis [37]. Furthermore, antibiotic treatment in at-risk patients may eliminate a bacterium that keeps mucormycosis under control. Coculturing *R. microsporus* and *Pseudomonas aeruginosa* resulted in the inhibition of spore germination via the secretion of bacterial siderophores [38]. The iron complex in bacterial pyoverdine has a much higher stability constant than that of hydroxamate RHF, indicating a better armament for intramicrobiome combat [39].

Motivated by the lack of a new tool for the noninvasive diagnosis of mucormycosis, we present the metabolome of a *R. microsporus* patient isolate, which was retrieved in 2019. This in vitro metabolic profile was compared to the urinal profile of the same patient with proven *R. microsporus* invasive infection confirmed by internal transcribed spacer (ITS) sequence-based identification. A new chromatographic method for carboxylate siderophore separation was developed, and the *Rhizopus* metabolome was probed by our software tool called CycloBranch [40].

## 2. Materials and Methods

### 2.1. Strain Cultivation, Fungal Extraction, and Metabolite Analysis

*R. microsporus* isolate was obtained from the sputum of an immunocompromised patient and grown in iron-depleted mineral medium (residual content 235.5 µg of Fe/L) containing 55 mM glucose, 50 mM NH_4_Cl, 11.2 mM KH_2_PO_4_, 7 mM KCl, 2.1 mM MgSO_4_ × 7H_2_O, 68 µM CaCl_2_ × 2H_2_O, and 20 µM ZnSO_4_ × 7H_2_O (pH 6.5). The medium was inoculated with spore suspension (10^7^ spores/mL) and incubated for 48 h at 30 °C with shaking (190 rpm). The supernatant fluid from culture was separated by centrifugation (5,000× *g*, 4 °C, 15 min) and subsequently lyophilized. Then, two-step liquid-liquid extraction was performed, similar to our previous methodology [27].

Briefly, the lyophilized sample was redissolved in water, extracted twice with ethyl acetate and dried under reduced pressure. The remaining aqueous phase was mixed with four equivalents of methanol and deep-frozen (−80 °C, 1 h). Precipitated proteins were removed by centrifugation (14,000× *g*, 4 °C, 10 min), and the supernatant was transferred to a vial with the residue from the evaporated ethyl acetate fraction and concentrated under reduced pressure. The pooled extract was resuspended in 5% liquid chromatography-mass spectrometry (LC-MS)-grade acetonitrile and injected onto an Acquity HSS T3 C18 analytical column (1.8 μm, 1.0 × 150 mm, Waters, Milford, MA, USA). Analytes were gradient-eluted with a 50 μL/min flow rate (A: 1% ACN with 0.1% formic acid in water, B: 95% ACN with 0.1% aqueous formic acid): 0 min, 2%; 2 min, 2%; 9 min, 60%; 11.0 min, 99%; 14 min, 99%; 14.5 min, 2%; and 20 min, 2% of B.

To quantify metabolites from crude extracts, we performed high-performance liquid chromatography (HPLC)-MS on a Dionex UltiMate 3000 UHPLC system (Thermo Fisher Scientific, Waltham, MA, USA) connected to a SolariX 12T Fourier transform ion cyclotron resonance mass spectrometer (Bruker Daltonics, Billerica, MA, USA) in electrospray ionization (ESI) positive-ion mode. The two CASI^TM^ (continuous accumulation of selected ions) windows were adjusted by a quadrupolar filter to 200–700 (low mass, LM) and 500–1500 (high mass, HM) Daltons. Qualitative and quantitative data processing was performed by our inhouse CycloBranch [40] version 2.0.8 and Bruker Data Analysis 5.0 software, respectively.

RHF and bis-imido-RHF were quantified in fermentation broths by the standard addition method. The standards were obtained from EMC Microcollections, GmbH (Tubingen, Germany). The limit of detection (LOD) and limit of quantitation (LOQ) were defined as the sum of the background average with 3 and 10 multiples of standard deviation, respectively. All samples were measured in triplicate.

### 2.2. Molecular Identification and Human Sample Collection

Template DNA was isolated from mycelium grown in yeast malt broth (Sigma–Aldrich, Ltd., St. Louis, MO, USA) for four days at 28 °C with shaking (100 rpm). The high-purity polymerase chain reaction (PCR) template preparation kit (Roche Diagnostics, Mannheim, Germany) was used as follows: mycelium was suspended in 10 mM EDTA buffer (1 mL), sonicated 3 × 20 s within a 2-min interval (5 W power, Microson ultrasonic disruptor, Labcaire) and centrifuged. For sample lysate, 200 µL of lysis buffer containing 0.2 M Tris-HCl (pH 7.5), 0.5 M NaCl, 0.01 M EDTA, 1% SDS, and 10 µL of lyticase (0.5 mg/mL) was added to the mycelial pellet and incubated at 37 °C for 30 min. The reaction was supplemented with 40 µL of proteinase K solution and incubated at 68 °C overnight. The purification further proceeded with the addition of 200 µL of binding buffer. PCR was carried out to amplify the fungal DNA parts of the 18S rRNA, ITS1 and ITS2, 5.8 S, and 28 S gene regions with the primer set F1422/ITS7. PCR amplicons were purified with the Roche purification kit, sequenced with the PCR and internal primers ITS1F, ITS2, and ITS4, analyzed by Lasergene (DNASTAR Inc., Madison, WI, USA) and queried in the International Society of Human and Animal Mycology ITS reference DNA barcoding database.

The urine and serum were obtained from a patient with an invasive *R. microsporus* infection. In line with the EORTC/MSG 2019 (European Organization for Research and Treatment of Cancer and the Mycoses Study Group) [41] criteria, mucormycosis was proven by PCR and rDNA sequencing. The specimens were collected twice between 2019 and 2020 at the Faculty Hospital in Prague Motol, Czech Republic. The study adhered to the Declaration of Helsinki, 2013, Good Clinical Practice, and was approved by the Ethics Committee for Multi-Centric Clinical Trials of the Charles University Hospital Motol, Prague (EK-826/19).

## 3. Results

### 3.1. HPLC Separation and Annotation of Rhizoferrin Analogs

The initial separation attempts started with the application of a standard Waters Acquity HSS T3 (1.0 × 150 mm, 1.8 μm) column. The stationary phase revealed considerable adsorption of desferri-RHF. Injections exceeding 1 μg/mL became visible as an iron complex of RHF when Fe^3+^ ions from HPLC stainless steel tubing were trapped by RHF. The desolvation temperature and ion transfer optical parameters were optimized to reduce the in-source fragmentation.

*R. microsporus* cultivation under iron-limited conditions and D-glucose as a carbon source provided RHF and imido-RHF as two dominant products in the small molecule CASI LM window (Figure 1A). Five months of storage at room temperature, even in a dried lyophilized state, caused extensive RHF decomposition to the imido- and bis-imido-RHF forms (Figure 1B). The RHF trace was visible with a deteriorated peak shape of the protonated molecules at m/z 437.140 (Figure 1E). The citryl-RHF intermediate eluted at 2.36 min and was characterized by the accurate mass of the protonated molecule at m/z 263.124 (263.124 calculated for C_10_H_18_N_2_O_6_).

For the first time, we report the production of glycosylated RHF analogs in directed biosynthesis (Figure 1G–I). Compared to their parent compounds (RHF and imido-RHF), both di- and tetra-Hex-RHFs had a mutually different and slightly lower affinity to the stationary phase. The positive-ion ESI mass spectra provided clusters of protonated and cationized species (data not shown). Ionic signals of the putative mono- and tri-Hex-RHFs coeluted with the corresponding di- and tetra-analogs. Although the formation of mono- and tri-Hex-RHFs cannot be excluded, at present, we conclude that the corresponding ionic species represent fragments rather than biosynthetic intermediates. On the other hand, di-Hex-imido-RHF was separated (Figure 1I).

The product ion mass spectra of the di- and tetra-Hex analogs revealed the consecutive losses of Hex units from the protonated molecules (Figure 1L,M) and the intrinsic fragmentations in the aglycone RHF part (Figure 1J,K). The ion compositions derived from exact mass measurements (Appendix A) do not contradict the fragmentation behavior indicated in Figure 1. CycloBranch provided the direct annotation of the components in the library search against the list of metabolites (Table 1).

### 3.2. Quantitation of Rhizoferrins

The fresh sample from the fermentation broth lyophilizate contained 345.3 ± 13.5, 1.2 ± 0.03, and 0.03 ± 0.002 mg/g of RHF, imido-RHF, and bis-imido-RHF, respectively. The LOD and LOQ calculated from the bis-imido-RHF signal-to-noise ratios were 1.0 and 3.2 ng/mL, respectively. The accurate LOD and LOQ for RHF could not be determined due to extreme sorption during HPLC separation. The complete standard addition procedure is described in the Appendix A. We can only estimate that a minimum RHF concentration of 1 µg/mL is needed for RHF recovery from the HSS T3 column. This concentration is two to three orders of magnitude higher than that expected for the RHF concentration in clinical samples.

## 4. Discussion

Aging of the lyophilizate exhibited considerable RHF and imido-RHF instabilities (Figure 1B). Due to enzymatic processes in human serum or urine, the RHF stability in those matrices is expected to be even more compromised. Hence, bis-imido-RHF or its metabolic forms were one of the few *R. microsporus* biomarkers remaining in bodily fluids for targeted metabolomics.

RHF’s glycosylation provides products that are more polar and possibly more enzymatically stable. We can only speculate whether RHF glycosylation may represent some benefit to the producer; e.g., biotransformation has been reported for detoxification of compounds. For example, *R. microsporus* converted zearalenone to its more soluble 4-β-D-glucopyranoside [42]. Of note, the presence of glucosyltransferases in the genus *Rhizopus* is not scarce, and these enzymes are used, e.g., in estrogen biotransformation [43].

RHF itself is an analytically difficult molecule for direct detection in patients with invasive mucormycoses, and its metabolic fate in the host body remains obscure. Our detection experiments with RHF analogs directly in human urine or serum samples have not yet been conclusive. The antifungal treatment likely stopped siderophore production before we could perform the actual sampling. Although RHF is the dominant product of *R. microsporus* cultivation under iron-restricted conditions, the RHF potential for zygomycete diagnostics needs to be demonstrated in the future with different sample preparation protocols, including chemical derivatization.

The mass spectrometer itself is sufficiently sensitive, as documented on urine analysis of the same patient, from which the *Rhizopus* isolate was originally obtained. The patient (M, 67) suffered from pulmonary mucormycosis, confirmed by microscopy, cultivation and PCR. The first urinal sampling started three months after initiating Noxafil antifungal treatment (Figure 2). Posaconazole (C_37_H_42_N_8_O_4_F_2_) is primarily metabolized via UDP glucuronosyltransferase [44]. In addition to the parent antifungal drug, we can directly monitor all its major metabolites, including hydroxyderivatives or glucuronides, in a single run.

Future studies are needed to determine the biotransformations of RHF in the host and to precisely reveal the chemical stability and protein binding of RHF. Patient specimen collection before the initiation of antifungal treatment is a prerequisite for successful detection of any iron-laden RHF analogs indicating mucoromycete proliferation in the host. This research is worth undertaking as it could provide noninvasive access to early biomarkers for a disease that provides blood cultures that are rarely positive, even when dissemination has occurred [45]. Our future attempts will be directed to RHF desoxy derivatives and chemical modification of rhizoferrin.

The comparative genomics approaches are also needed for the allocation of genes coding the putative glucosyltransferases. The whole-genome shotgun sequencing predicted glycosyltransferase/glycogen phosphorylase genes, e.g., in *R. microsporus* ATCC strains 52,813, 11,559, 52,814, 62,417, one CBS-344.29 strain, and in *R. delemar* RA 99-880 (http://fungi.ensembl.org). By May 2020 no evidence on rhizoferrin, its glycosylated forms, or other analogs have been reported in the literature either in the patients’ samples or in the animal models. We do not know whether the RHF is truly secreted in the human host at the invasive stage of the disease. Similarly, no information is available about RHF’s possible renal secretion and protein binding. As siderophore secretion has been defined as one of the most important factors of virulence, including the RHF secretion [46], we hypothesize that the obstacles with RHF detection in the host rise rather from its high reactivity and instability. It is, therefore, our closest plan to set up an animal infection model with ^68^Ga-labeled RHF as a radiotracer. 

## Figures and Tables

**Figure 1 jof-06-00089-f001:**
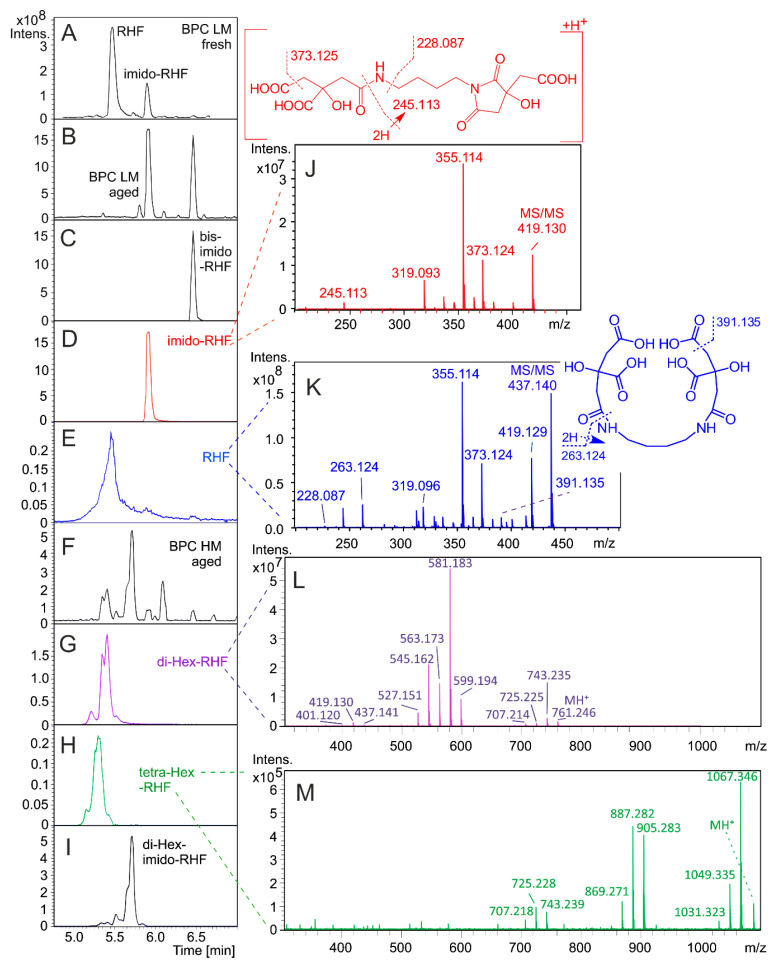
Chromatographic separation and tandem mass spectrometry of *R. microsporus* secondary metabolites. BPC: base peak chromatogram (CASI LM or HM windows), Hex stands for hexose. (**A**,**B**), BPC in the 200–700 Daltons mass range run on fresh or aged (5 months) lyophilizates; (**C**–**E**), reconstructed ion mass chromatograms corresponding to bis-imido-RHF, imido-RHF, and RHF, respectively (0.005 Dalton window). The BPC and selected ion chromatograms of di-Hex-RHF, tetra-Hex-RHF, and di-Hex-imido-RHF are shown in panels (**F**–**I**), respectively (all HM). Panels (**J**–**M**) refer to the product ion mass spectra (10 eV collisional energy) of imido-RHF, RHF, di-Hex-RHF, and tetra-Hex-RHF, respectively. The structure on the top indicates the intrinsic fragmentation in the RHF molecule.

**Figure 2 jof-06-00089-f002:**
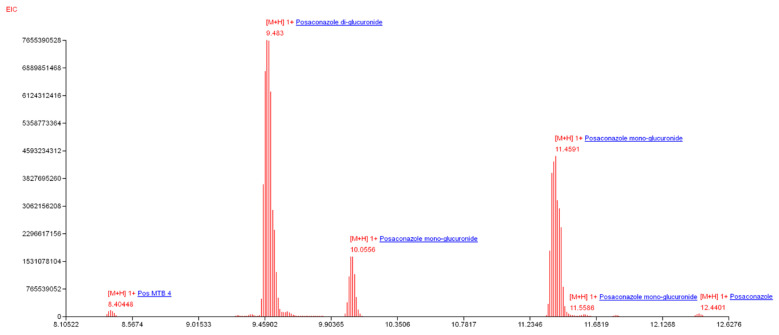
Annotation of posaconazole and its metabolites in the patient’s urine by CycloBranch (15 ppm accuracy, 0.4% minimum relative intensity threshold, 1,000,000 minimum absolute intensity threshold, full pattern with a minimum of three isotope peaks in at least three consecutive scans). The CycloBranch internal mucormycotic database was amended with current antifungals and their metabolites [36].

**Table 1 jof-06-00089-t001:** Small molecules described in Mucorales. If not stated otherwise, the abbreviated genera *M., R*., *L.,* and *C*. stand for *Mucor, Rhizopus, Lichtheimia*, and *Cunninghamella*, respectively. M: monoisotopic molecular weight; RHF: rhizoferrin; ^1^ Reported in *Phycomyces blakesleeanus*; ^2^ Described in *Blakeslea trispora*; ^3^ Known also as glomuferrin [8]; ^4^ Known as staphyloferrin; * This work.

M	Metabolite	Formula	*R. microsporus*	*M. irregularis*	*R. delemar*	*L. corymbifera*	*Rhizobium meliloti*	*C. echinulata*	*C. elegans*	*Rhizomucor pusillus*	*M. circinelloides*	References
119.0219	3-Nitropropionic acid	C_3_H_5_NO_4_										[5]
262.1165	RHF citryl intermediate	C_10_H_18_N_2_O_6_										[9]
289.1804	Trisporic acid E^1^	C_18_H_25_O_3_										[10]
290.1882	Trisporic acid A	C_18_H_26_O_3_										[11]
304.1675	Trisporic acid B	C_18_H_24_O_4_										[11]
306.1831	Trisporic acid C	C_18_H_26_O_4_										[11]
320.1624	Trisporic acid D^2^	C_18_H_24_O_5_										[12]
376.1720	Rhizobactin	C_15_H_26_N_3_O_8_										[13]
384.1169	Desoxy-bis-imido-RHF	C_16_H_20_N_2_O_9_										[14]
400.1118	Bis-imido-RHF ^3^	C_16_H_20_N_2_O_10_										[14,15]
404.1431	Di-desoxy-RHF	C_16_H_24_N_2_O_10_										[14]
418.1224	Imido-RHF	C_16_H_22_N_2_O_11_										[14,15]
420.1380	Mono-desoxy-RHF	C_16_H_24_N_2_O_11_										[14]
422.1173	Nor-RHF ^4^	C_15_H_22_N_2_O_12_										[14]
432.1744	Di-methyl-di-desoxy-RHF	C_18_H_28_N_2_O_10_										[14]
434.1537	Methyl-desoxy-RHF	C_17_H_26_N_2_O_11_										[14]
435.2404	Rhizovarin F	C_27_H_33_NO_4_										[16]
436.1329	Rhizoferrin (RHF)	C_16_H_24_N_2_O_12_										[9,14]
450.1122	2-Oxo-RHF	C_16_H_22_N_2_O_13_										[14]
450.1486	Homo-RHF	C_17_H_26_N_2_O_12_										[14]
452.1278	2-Oxa-homo-RHF	C_16_H_24_N_2_O_13_										[14]
464.1642	2-Methyl-homo-RHF	C_18_H_28_N_2_O_12_										[14]
603.3560	Rhizovarin D	C_37_H_49_NO_6_										[16]
615.3560	Rhizovarin E	C_38_H_49_NO_6_										[16]
645.3302	Rhizovarin C	C_38_H_47_NO_8_										[16]
769.2111	Rhizovarin A	C_37_H_44_ClNO_8_										[16]
783.2268	Rhizovarin B	C_38_H_46_ClNO_8_										[16]
742.2280	Di-Hex-imido-RHF	C_28_H_42_N_2_O_21_										*
760.2386	Di-Hex-RHF	C_28_H_44_N_2_O_22_										*
1084.3442	Tetra-Hex-RHF	C_40_H_64_N_2_O_32_										*

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
