# Peer review of "Rhizoferrin Glycosylation in Rhizopus microsporus"

_jof, 2020, doi:10.3390/jof6020089_

Round 1

Reviewer 1 Report

Skriba et al. are presenting very interesting work on glycosylation of the siderophore rhizoferrin (RHF) in a clinical Rhizopus microsporus isolate. The manuscript is well written, and I have only minor requests which are detailed below. 

  1. Line 13: a mix up of words
  2. Table 1: please spell out RHF in the legend
  3. Whereas it is clearly possible to provoke RHF glycosylation in vitro, there so far seems to be no evidence of glycosylated RHF in patient samples. The authors introduce human samples collection in the Materials and Methods part, and discuss sensitivity issues that prevented detection of (glycosylated) RHF in these samples. I would like to see a para in the Results part summarizing their findings.
  4. Did the authors try and detect (glycosylated) RHF in samples from patients before treatment?

Author Response

Dear Editor:

In parallel to the original submission, we sent the manuscript to the Springer Nature Author Services for English editing. The certificate we uploaded to the MDPI portal now and continued the editing by addressing criticisms that arose from the reviewer’s reports. All these criticisms were fair and correct. Also, we made a couple of our own edits and updated literature in press to that with the pages. Please, express our gratitude to all three reviewers.

With kind regards,

Vladimir Havlicek

Reviewer No. 1

We thank this reviewer for careful reading and the overall positive note of the review. We addressed all four comments.

  1. Line 13: a mix up of words

Response: We thank this reviewer and corrected the sentence.

  1. Table 1: please spell out RHF in the legend

Response: The rhizoferrin was spelled out.

  1. Whereas it is clearly possible to provoke RHF glycosylation in vitro, there so far seems to be no evidence of glycosylated RHF in patient samples. The authors introduce human samples collection in the Materials and Methods part, and discuss sensitivity issues that prevented detection of (glycosylated) RHF in these samples. I would like to see a para in the Results part summarizing their findings.

Response: The reviewer is right, and we amended the following text to the Results paragraph: “By May 2020 no evidence on rhizoferrin, its glycosylated forms or other analogs have been reported in the literature either in the patients’ samples or in the animal models. We do not know whether the RHF is truly secreted in the human host at the invasive stage of the disease. Similarly, no information is available about RHF’s possible renal secretion and protein binding. As siderophore secretion has been defined as one of the most important factors of virulence, including the RHF secretion (https://doi.org/10.3389/fcimb.2017.00107), we hypothesize that the obstacles with RHF detection in the host rise rather from its high reactivity and instability. It is, therefore, our closest plan to set up an animal infection model with Ga-68-labelled RHF as a radiotracer.”   

  1. Did the authors try and detect (glycosylated) RHF in samples from patients before treatment?

Response: In two years, we have collected six patients. Most of them were on preemptive prophylaxis or in intensive antifungal therapy. In one of the patients, we saw the putative RHF glycoside (retention time, the exact mass of the protonated molecule), but the precise masses of the fragments did not match. In another case, we detected the ferri-form of a putative desoxy-RHF (exact mass, one ppm accuracy), but we did not have the corresponding standard. Therefore, we do not consider our results as conclusive unless we have the complete evidence (standard retention time, the exact mass of the parent, exact masses of the fragments in agreement with those of the standard). Out of these six patients (some patients were sampled more than once), only two samples were collected before the antifungal treatment. Unfortunately, even these two samples were negative for RHF analogues.

Reviewer 2 Report

Rhizoferrin, imido-rhizoferrin and bis-imido-rhizoferrin were initially identified and characterized in R. microspores by the group of Günther Winkelmann. This should be acknowledged in the text!

Drechsel et al. (1991) Rhizoferrin — a novel siderophore from the fungusRhizopus microsporus var.rhizopodiformis.Biology of Metals volume 4, pages238–243(1991)

Therefore, the identification of this siderophore is not surprising. The novelty of this study is the identification of the glycosylated forms. However, these are insufficiently described. Please describe the evidences clearer. Are these degradation products? Where are the hexose groups conjugated – please provide rough structures? Do these rhizoferrin forms still chelate iron?

Line 182: only amounts of RHF, imido-RHF and bis-imido-RHF are given. What about the glycosylated forms?

Lines 156-158: Why wasn´t Rhizoferrin iron saturated before analysis to reach homogenous analytes?

Lines 213-219: This part is confusing as nothing is mentioned in the Results section. Is this ”data not shown”? The part concerning the detection in humans samples is immature: it has not been tested if rhizoferrin is renally excreted; also protein-binding has not been examined. This could be examined preclinically in animal models.

Minor:

Line 13: relieFcerebs? relies

Line 54: I do not think that rhizoferrin has been proven to be important for virulence (also not disproven)

Lines 60-61: the primary function of siderophores is chelation of iron for acquisition and/or intracellular handling of iron.

Line 63: what is passivation

Line 65: therapeutics: please specify, also include desferal (against iron and aluminum overload). I think that only desferal and cefiderocol (a cephalosporin-siderophore conjugate) are commercially applied siderophore/siderophore derivatives. All other examples are in preclinical phase.

Line 67: regarding diagnostic potential, imaging of fungal and bacterial infections (Aspergillus fumigatus and Pseudomonas aeruginosa) should be mentioned. With respect to the use of siderophores as biomarkers, I am surprised that the promising studies with clinical patient samples are not mentioned:

Orasch T, Prattes J, Faserl K, et al. Bronchoalveolar lavage triacetylfusarinine C (TAFC) determination for diagnosis of invasive pulmonary aspergillosis in patients with hematological malignancies. J Infect. 2017;75(4):370‐373. doi:10.1016/j.jinf.2017.05.014

Orasch T, Prattes J, Faserl K, et al. Bronchoalveolar lavage triacetylfusarinine C (TAFC) determination for diagnosis of invasive pulmonary aspergillosis in patients with hematological malignancies. J Infect. 2017;75(4):370‐373. doi:10.1016/j.jinf.2017.05.014

Line71: I wouldn´t call lipocalins (1 and 2 = siderocalins) siderophores.

Line 83: ”activity” instead of ”sensitivity”

Line 109: how much methanol; is rhizoferrin in the ethylacetate or water phase.

Line 167: ”glycated”: for consistency, I suggest to use glycosylated throughout the whole manuscript.

Line 210: what would be the rationale for detoxification? Rhizoferrin is not toxic for the producer!

There are more specific reviews with regard to siderophores than (20).

(22) and (36) appear to be inappropriate references. In general, I suggest to reference preferentially original publications and not secondary literature.

Author Response

Reviewer No. 2

Dear Editor:

In parallel to the original submission, we sent the manuscript to the Springer Nature Author Services for English editing. The certificate we uploaded to the MDPI portal now and continued the editing by addressing criticisms that arose from the reviewer’s reports. All these criticisms were fair and correct. Also, we made a couple of our own edits and updated literature in press to that with the pages. Please, express our gratitude to all three reviewers.

With kind regards,

Vladimir Havlicek

We would like to sincerely thank specially to this second reviewer, as he correctly addressed most of the critical issues we have been coping with within the past couple of weeks. All comments and criticisms were fair and addressed.

  1. Rhizoferrin, imido-rhizoferrin, and bis-imido-rhizoferrin were initially identified and characterized in R. microspores by the group of Günther Winkelmann. This should be acknowledged in the text! Drechsel et al. (1991) Rhizoferrin — a novel siderophore from the fungusRhizopus microsporusrhizopodiformis.Biology of Metalsvolume 4, pages238–243(1991).

Response: We have cited five G. A. Winkelmann’s references, and we agree with this reviewer and also added this sixth one (now appearing as No. 18).

  1. Therefore, the identification of this siderophore is not surprising. The novelty of this study is the identification of the glycosylated forms. However, these are insufficiently described. Please describe the evidences clearer. Are these degradation products? Where are the hexose groups conjugated – please provide rough structures? Do these rhizoferrin forms still chelate iron?

Response: We again agree with this reviewer. We do not think that these are degradation products. Our working hypothesis is that the RHF glycosides arose from the directed biosynthesis as the cultivation was performed on glucose. Our attempt to grow R. microsporus on glycerol failed; no pellets were formed in four days. We plan a larger-scale (2L) cultivation and will isolate the glycosides in enough quantity and purity needed for NMR spectroscopy. Only with pure and authenticated compounds, the complexation studies can be carried out. Within the ten-day response window given by the publisher, we could not make it due to a timely manner.

  1. Line 182: only amounts of RHF, imido-RHF, and bis-imido-RHF are given. What about the glycosylated forms?

Response: The glycosylated forms have different solubility and higher polarity compared to the parent RHF. Very likely, they also will have different ionization efficiencies, and droplets in the electrospray will have different surface tensions compared to the parent RHF. It would not be fair to consider that the RHF and glyco-RHFs will have the same response factors. This information can be provided once we have purified standards.

  1. Lines 156-158: Why wasn´t Rhizoferrin iron saturated before analysis to reach homogenous analytes?

Response: We are thankful for this comment, and frankly, we have tested the saturation both by FeSO4, Fe-citrate, and FeCl3. In our hands, the responses of the ferri-RHF were not more sensitive than those of the desferri-RHF forms we tested on two HPLC columns. The first one was Waters Acquity HSS T3 C18 analytical column, 1.8 μm, 1.0 × 150 mm, and this column we also used for reporting in the paper. Also, we worked with PEEK InertSustain AQ-C18 (1.9 μm, 2.1 × 150 mm, GL Sciences, Japan), both in isocratic or gradient conditions, but the sensitivity or separation power, we achieved, was not better. Instead of iron saturation, we think about carboxylate derivatization to achieve nanogram/mL sensitivity we need for clinical samples.

  1. Lines 213-219: This part is confusing as nothing is mentioned in the Results section. Is this ”data not shown”? The part concerning the detection in humans samples is immature: it has not been tested if rhizoferrin is renally excreted; also protein-binding has not been examined. This could be examined preclinically in animal models.

Response: We agree with this reviewer and amended the following text to the Results paragraph (including a new reference 44): “By May 2020 no evidence on rhizoferrin, its glycosylated forms or other analogs have been reported in the literature either in the patients’ samples or in the animal models. We do not know whether the RHF is truly secreted in the human host at the invasive stage of the disease. Similarly, no information is available about RHF’s possible renal secretion and protein binding. As siderophore secretion has been defined as one of the most important factors of virulence, including the RHF secretion [44], we hypothesize that the obstacles with RHF detection in the host rise rather from its high reactivity and instability. It is, therefore, our closest plan to set up an animal infection model with Ga-68-labelled RHF as a radiotracer.”   

  1. Minor: Line 13: relieFcerebs? Relies

Response: Thank you, corrected.

  1. Line 54: I do not think that rhizoferrin has been proven to be important for virulence (also not disproven)

Response: We added a review reference to the main text (Iron and Virulence in Francisella tularensis. Front Cell Infect Microbiol 2017, 7, article No 107).

  1. Lines 60-61: the primary function of siderophores is chelation of iron for acquisition and/or intracellular handling of iron.

Response: Yes, indeed, every author starts with such a claim. We wanted to be different and picked up another piece of information first for the newcomers to the field. So, we started with generally less familiar applications and iron complexation we mentioned later on.

  1. Line 63: what is passivation

Response: Passivation is the opposite process of nutritional immunity. We referenced this function with yersiniabactin at a high copper concentration (see e.g., Henderson’s work: Metal Selectivity by the Virulence-Associated Yersiniabactin Metallophore System, DOI: 10.1074/jbc.RA118.004483).

  1. Line 65: therapeutics: please specify, also include desferal (against iron and aluminum overload). I think that only desferal and cefiderocol (a cephalosporin-siderophore conjugate) are commercially applied siderophore/siderophore derivatives. All other examples are in preclinical phase.

Response: Thank you for the comment. Desferal (beta-thalassemia treatment and aluminium overload, McCarthy JT, Milliner DS, Johnson WJ. Clinical experience with desferrioxamine in dialysis patients with aluminium toxicity. Q J Med. 1990;74(275):257‐276) and cefiderocol (Cefiderocol: A Novel Siderophore Cephalosporin Defeating Carbapenem-resistant Pathogens, https://doi.org/10.1093/cid/ciz823) applications have been amended. One inappropriate reference was removed.

  1. Line 67: regarding diagnostic potential, imaging of fungal and bacterial infections (Aspergillus fumigatus and Pseudomonas aeruginosa) should be mentioned. With respect to the use of siderophores as biomarkers, I am surprised that the promising studies with clinical patient samples are not mentioned: Orasch T, Prattes J, Faserl K, et al. Bronchoalveolar lavage triacetylfusarinine C (TAFC) determination for diagnosis of invasive pulmonary aspergillosis in patients with hematological malignancies. J Infect. 2017;75(4):370‐373. doi:10.1016/j.jinf.2017.05.014

Response: We again agree with this reviewer and added a newer reference with urinal TAFC (the same author group: Hoenigl, M.; Orasch, T.; Faserl, K.; Prattes, J.; Loeffler, J.; Springer, J.; Gsaller, F.; Reischies, F.; Duettmann, W.; Raggam, R.B., et al. Triacetylfusarinine C: A urine biomarker for diagnosis of invasive aspergillosis. J Infect 2019, 78, 150-157).

  1. Line71: I wouldn´t call lipocalins (1 and 2 = siderocalins) siderophores.

Response: We agree with this reviewer and deleted the controversial protein claim.

  1. Line 83: ”activity” instead of ”sensitivity”

Response: We changed the term as desired.

  1. Line 109: how much methanol; is rhizoferrin in the ethyl acetate or water phase.

Response: The RHF is hydrophilic and exclusively goes to aqueous phase. As we had no clue about the RHF decomposition products or metabolism, we used an analytically safer all-in-one protocol including the ethyl acetate extra step. We have used the four equivalents of methanol, as now stated in the main text.

  1. Line 167: ”glycated”: for consistency, I suggest to use glycosylated throughout the whole manuscript.

Response: the term “glycosylated” is now used throughout the whole manuscript.

  1. Line 210: what would be the rationale for detoxification? Rhizoferrin is not toxic for the producer!

Response: The current literature indicates that glycosylation can be a detoxifying process either in microbial combat (Detection of modified mycotoxin D3G in a dual culture of Trichoderma and Fusarium indicated that detoxification via glycosylation may not be exclusive to plants. 10.3390/toxins8110335) or in escaping the immune system of the host (DOI: 10.1039/c4mt00333k). We do not know, if the glycosylation represents some advantage to the producer during R. microsporus cultivation in vitro. As we agree with the reviewer, we softened our wording by “may” – see the revision.

  1. There are more specific reviews with regard to siderophores than (20).

Response: Thank you for the comment. We switched the reference (20) to Hider and Kong.

  1. (22, Metabolites) and (36, ChemListy) appear to be inappropriate references. In general, I suggest to reference preferentially original publications and not secondary literature.

Response: We again agree with this reviewer. We removed Metabolites and changed ChemListy to Inorganic Chemistry 2006.

Reviewer 3 Report

The manuscript “Rhizoferrin glycosylation in Rhizopus microsporus” by Anton Skriba and coworkers describes a screen of derivatives of rhizoferrin in extracts from mycelial culture and a urine sample. The Rhizopus strain was isolated from a patient with pulmonary mucormycosis. The samples were analysed in HPLC and FTICR. The aims of the study are clearly stated as well as the methods involved are state of art. Analysis of Rhizopus microsporus metabolites revealed the presence of glycosylated RHF forms. 

The study showed the instability of diverse RHS derivatives even in a lyophilized sample and the necessity of urine sample collection prior to antifungal treatment. 

Unfortunately, these derivatives were not detected in the urine sample. Authors suggested further possible approaches which will depend obviously on the availability of patient samples.

Despite negative results, this study brings some hope for faster mucormycosis diagnostics.

It would be recommendable to hypothesise about glucosyltransferases responsible for the identified modification eg. based on comparative genomics.

Are the authors planning the reveal the chemical structure and properties of the new compounds?

Author Response

Reviewer No. 3

Dear Editor:

In parallel to the original submission, we sent the manuscript to the Springer Nature Author Services for English editing. The certificate we uploaded to the MDPI portal now and continued the editing by addressing criticisms that arose from the reviewer’s reports. All these criticisms were fair and correct. Also, we made a couple of our own edits and updated literature in press to that with the pages. Please, express our gratitude to all three reviewers.

With kind regards,

Vladimir Havlicek

We wish to thank this reviewer for the generally positive note, and we responded to both queries:

It would be recommendable to hypothesize about glucosyltransferases responsible for the identified modification e.g. based on comparative genomics.

Response: We thank for this comment, and the following wording was amended into the terminal part of the text: “The comparative genomics approaches are also needed for the allocation of genes coding the putative glucosyltransferases. The whole-genome shotgun sequencing predicted glycosyltransferase/glycogen phosphorylase genes, e.g., in R. microsporus ATCC strains 52813, 11559, 52814, 62417, one CBS-344.29 strain, and in R. delemar RA 99-880 (http://fungi.ensembl.org).”

Are the authors planning the reveal the chemical structure and properties of the new compounds?

Response: Yes, it is our plan. First, we have started with a more manageable task, which is the large-scale fermentation to get enough glycosyl-RHF needed for NMR spectroscopy. We have also convinced our veterinarians to begin with the experimental infections in rats. Ga-68 labeling is now mentioned in the revised text. Without an animal model, it would be almost impossible to learn about the RHF renal secretion during the invasive mucormycosis in a host. 

Round 2

Reviewer 2 Report

The authors addressed most issues raised and considerably improved the manuscript.

However, I still have two issues

Lines 55 and and 263 (orginal point 7.): This article and the respective sentences deal with Mucorales/Rhizopus, while the cited article regarding the role of Rhizoferrin in virulence deals with the bacterium Francisella. This is misleading. It should be clearly stated in the text, that the role in virulence has been shown in the bacterium Francisella. A role in virulence of Rhizoferrin has not been shown in Mucorales species so far.  

Line 218 (oriningal point 16.): ”detoxification” still sounds weird as no toxic function of rhizoferrin has been reported for the producer. I suggest to reword the text, e.g., ”......benefit to the producer, e.g. biotransformation has been reported for detoxification of compounds.”

Author Response

Response:

Thank you for this important comment, the original wording was misleading, indeed. We reformulated the sentence and placed a correct reference.

Mucorales also synthesize polycarboxylate siderophores, including rhizoferrin (RHF) [18], which have a much weaker binding activity than hydroxamate siderophores but may be important for microbial virulence, as reported, e.g., in Francisella tularensis [19].

Response:

We again thank this reviewer for suggesting much better wording. Instead of the original

“We can only speculate whether RHF glycosylation may represent some benefit to the producer, e.g., in the detoxification process.”

we now have the construct suggested by this reviewer:

“We can only speculate whether RHF glycosylation may represent some benefit to the producer; e.g., biotransformation has been reported for detoxification of compounds.”
